# Evidence for continuity of interstitial spaces across tissue and organ boundaries in humans

Odise Cenaj[1], Douglas H. R. Allison [1], Rami Imam[1], Briana Zeck[1], Lilly M. Drohan [1], Luis Chiriboga [1], Jessica Llewellyn[2], Cheng Z. Liu[1], Young Nyun Park [3], Rebecca G. Wells [2,4,5,6,7] & Neil D. Theise [1,7✉]

Bodies have continuous reticular networks, comprising collagens, elastin, glycosaminoglycans, and other extracellular matrix components, through all tissues and organs. Fibrous coverings of nerves and blood vessels create structural continuity beyond organ boundaries. We recently validated fluid flow through human fibrous tissues, though whether these interstitial spaces are continuous through the body or discontinuous, confined within individual organs, remains unclear. Here we show evidence for continuity of interstitial spaces using two approaches. Non-biological particles (tattoo pigment, colloidal silver) were tracked within colon and skin interstitial spaces and into adjacent fascia. Hyaluronic acid, a macromolecular component of interstitial spaces, was also visualized. Both techniques demonstrate interstitial continuity within and between organs including within perineurium and vascular adventitia traversing organs and the spaces between them. We suggest that there is a body-wide network of fluid-filled interstitial spaces that has significant implications for molecular signaling, cell trafficking, and the spread of malignant and infectious disease.

[1] Department of Pathology, New York University Grossman School of Medicine, New York, NY, USA. [2] Divison of Gastroenterology and Hepatology, Department of Medicine, Perelman School of Medicine at the University of Pennsylvania, Philadelphia, PA, USA. [3] Department of Pathology, Department of Pathology, Yonsei University College of Medicine, Seoul, South Korea. [4] Department of Bioengineering, School of Engineering and Applied Sciences, The University of Pennsylvania, Philadelphia, PA, USA. [5] Department of Pathology and Laboratory Medicine, Perelman School of Medicine at the University of Pennsylvania, Philadelphia, PA, USA. [6] Center for Engineering MechanoBiology, The University of Pennsylvania, Philadelphia, PA, USA. [7] These authors contributed equally: Rebecca G. Wells, Neil D. Theise. ✉email: neil.theise@nyulangone.org

The work of Franklin Mall over a century ago[1,2], as well as modern-day decellularization techniques[3], demonstrate that there are "reticular networks" made up of collagens, elastin, glycosaminoglycans, and other extracellular matrices (ECM) components surrounding, within, and between organs. These networks have biological and mechanical roles in defining the architecture and physiology of organs and, as a result, are now used as scaffolding for the creation of customized organ grafts for regenerative medicine[4]. Multiorgan decellularization has further confirmed that ECM networks extend beyond the confines of single organs to involve neighboring structures, including thoracic (heart), abdominal (liver, gut, kidneys) and pelvic (uterus, prostate, urinary bladder) organs with their vasculature and surrounding fibrous adventitial sheaths, creating structural continuity across organ boundaries[5]. Continuity between mesenteric fascia and the connective tissues of the small intestinal and colonic walls has also been previously recognized[6–9]. More recently, the decellularization of entire fetal sheep shows that the connective tissue network is continuous throughout the body and that the connective tissue of nerves creates structural continuity between the nervous system and other tissues[3]. Dissection of human bodies likewise demonstrates continuity across large, multiorgan regions of the body, including the entirety of the dermis and the fascia of diverse organs and organ systems[10–13].

Clinical disciplines including osteopathy have suggested that these connective tissue networks contain fluid and represent a body-wide communications network, akin to interstitial spaces, although this lacks detailed microscopic confirmation. Interstitial spaces in which nutrient and waste exchange take place have historically been recognized at two scales: intercellular spaces (<1 micron) and pericapillary spaces (~10 microns)[14]. Studies have described fluid flow through large interstitial spaces of the human extrahepatic bile duct submucosa and the human dermis, 50–70 μm below the epithelial surface[15,16]. It has further been shown that other fibrous tissues, including the submucosae of all other visceral organs and the subcutaneous fascia, are structurally similar, and hypothesized that they likewise support fluid flow. In all of these tissues, the spaces were defined by a network of collagen bundles 20–70 μm in diameter. Many of the collagen bundles were lined by spindle-shaped cells that co-expressed vimentin and CD34, but were devoid of endothelial ultrastructural features and were thus considered fibroblast-like cells. In this context, we refer to them as "interstitial lining cells." In vivo endomicroscopy has shown that musculoskeletal fascia includes similar large-caliber fluid-filled spaces[17]. It remains unclear, however, whether these interstitial compartments are continuous through the body or represent discontinuous fluid-filled channels confined within individual organs. A limited demonstration of such intra-tissue continuity is found in the work of Mall, who reported that the interstitial spaces of liver portal tract stroma are continuous between peri-arterial, peri-venular, and periductal compartments (the "space of Mall")[1], but these studies were only at the local, intrahepatic level.

We have now investigated the question of continuity of interstitial spaces. Two orthogonal approaches were employed. The first was to study the movement of non-biological particles (tattoo pigment and colloidal silver) across tissue compartments within the colon and skin and into the adjacent fascia. The second examined the distribution of hyaluronic acid (HA), a macromolecular component of the smallest interstitial spaces (i.e., between cells and around capillaries) as well as the larger fibroconnective tissue spaces we recently identified[14,18–20].

## Results

### Tattoo pigment and colloidal silver are found distant from the original sites of application. Skin samples with tattoos injected

into the dermis for cosmetic reasons were examined for the presence of pigment particles distant from the dermis (Fig. 1). In three samples obtained from different patients, particles were identified in the papillary and reticular dermis and subcutaneous fascia (Fig. 1a). The particles were localized both intracellularly, within the cytoplasm of macrophages and interstitial lining cells, and extracellularly, within interstitial spaces between collagen bundles of the collagenous network of the dermis and subcutaneous fascia (Fig. 1b, c). Silver particles were observed in similar locations in two samples from a patient who developed argyria after topical application of colloidal silver (Supplementary Fig. 1a–f). Silver particles were also identified in the adnexa, perivascular adventitia, and perineurium in the dermis (Supplementary Fig. 1g–l).

Colon resection specimens with endoscopically injected tattoos also demonstrated pigment particles distant from the original submucosal injection site. In samples from all five specimens studied, pigment particles were identified not only in the colonic submucosa but also in the muscularis propria and mesenteric fascia (Fig. 2a). In a similar fashion to the findings in the skin, pigment particles were both intracellular and within interstitial spaces of the collagenous network of colonic submucosa, muscularis propria, and mesenteric fascia (Fig. 2b–d). We previously demonstrated such movement of tattoo pigment from colonic submucosa to draining lymph nodes of the mesentery[15].

Because particles are found within macrophages, which were shown previously to migrate to regional lymph nodes[15], one possible explanation for the appearance of extracellular particles at a distance is that they were carried there intracellularly and then released. Additionally, it is possible with the tattoo pigment that the initial injection was deep enough to explain our findings, though this would not explain the silver particles as they were absorbed after topical application. We, therefore, measured the diameter of the extracellular tattoo pigment particles as a function of the depth of their location in the bowel (Fig. 3a). Particles in deep mesenteric interstitial spaces were significantly smaller than those in more superficial compartments. Figure 3b shows the aggregated data for all five colon tattoo specimens in the three levels examined: submucosa, muscularis propria, mesenteric fascia. Fifty particles were measured in each layer of each case for a total of 750 particles. Data from each individual case is shown in Supplementary Fig. 2. Source data for these figures is shown in Supplementary Data 1. The same distribution of particles by size was seen in all specimens as is also reflected in the aggregate data. In aggregate, the mean particle size in deep mesenteric fascia was 0.41 vs 0.53 μm in the muscularis propria ($p = 3.1 \times 10^{-23}$), and 0.64 μm in the submucosa ($p = 7.2 \times 10^{-44}$ vs deep mesenteric fascia, $p = 4.5 \times 10^{-11}$ vs muscularis propria).

### Hyaluronic acid staining shows continuity between interstitial spaces across organ boundaries. HA is found in interstitial spaces throughout the body at all stages of development[14,18–20]. The physical properties of HA suggest that it regulates the flow of fluid and other solutes and small molecules within the interstitial fluid[21]; it did not prevent filling by fluorescein in vivo[15]. We confirmed by staining with HA binding protein (HABP) that it is found broadly in intercellular, pericapillary, and perineural, and submucosal and dermal interstitial spaces. This staining showed that non-vascular spaces that appear as white and therefore seeming empty by H&E staining in fact contain HA (Supplementary Fig. 3). The smallest interstitial spaces between cells are filled with HA including those between epidermal keratinocytes (Supplementary Fig. 3a, b) and within dermal nerve fibers (Supplementary Fig. 3g, h). Pericapillary-scale interstitial spaces

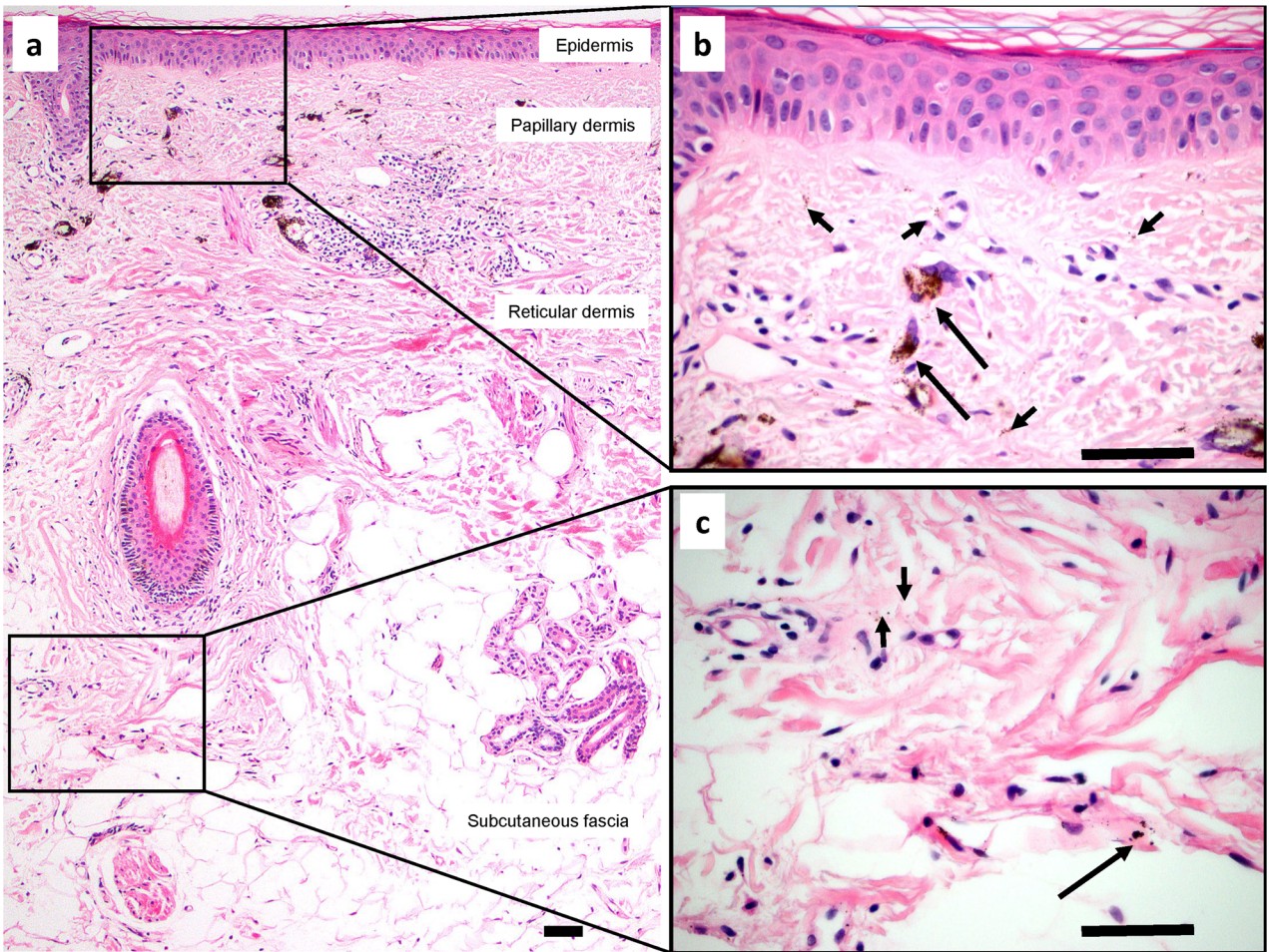

**Fig. 1 Tattoo pigment in interstitial spaces of the dermis and subcutaneous fascia. a** H&E section of skin and subcutaneous fascia with cosmetically injected brown-black tattoo pigment. Pigment particles are present in the papillary and reticular dermis and subcutaneous fascia, visible at low magnification. **b**, **c** Higher magnification views of the rectangular areas demonstrate both intracellular particles (within macrophages; long arrows) and extracellular particles (within interstitial spaces; short arrows) of the papillary dermis (**b**), reticular dermis, and subcutaneous fascia (**c**). Scale bars = 100 μm.

(lamina propria of the colon) were similar (Supplementary Fig. 3c, d). Larger, fibroconnective tissue interstitial spaces filled by HA were observed in the dermis (Supplementary Fig. 3a, b) and submucosa and in peri-arterial (Supplementary Fig. 3e, f) and perineurial fibroconnective tissues (Supplementary Fig. 3g, h). HA is thus a surrogate marker of many if not most interstitial spaces.

Staining for HA in the colon demonstrated that there were connections between all layers from lamina propria through muscularis mucosae to the submucosa, and then through muscularis propria into subserosa and mesenteric fascia (Fig. 4). Continuity between interstitial spaces in the submucosae and the perivascular adventitia and perineurium in the bowel wall was also evident (Supplementary Fig. 4).

In the skin, HA-filled interstitial spaces were continuous from intercellular spaces of the epidermis into the papillary dermis (Supplementary Fig. 3a, b and Fig. 5a, b) and then to the reticular dermis (Fig. 5a–d) and deeper into the subcutaneous fascia (Fig. 5a, b, e, f). Intercellular and pericapillary interstitial spaces of subcutaneous adipocytes are continuous with the interwoven subcutaneous fascia and with perivascular adventitia (Fig. 5e, f).

Similarly, the space of Mall in the portal tracts of the liver shows HA staining between the interstitial spaces of stroma around the intrahepatic bile duct and of the adventitial stroma around the hepatic artery and portal vein, indicating continuity (Supplementary Fig. 5).

**Multiple tumor types move through interstitial spaces**. Movement of the tumor through these interconnected interstitial spaces is demonstrated in cases of peribiliary spread of cholangiocarcinoma, which tracks between collagen fibers through the biliary submucosa and throughout the space of Mall while respecting the rigid boundary of the limiting plate of hepatocytes (Supplementary Fig. 6a, b)[22,23]. Similarly, colon adenocarcinoma may be seen moving between collagen bundles and between muscle bundles of the muscularis propria into the mesentery (Supplementary Fig. 6c, d) and malignant melanoma is known to give rise to "in transit" metastases within the dermis without evidence of lymphovascular invasion (Supplementary Fig. 6e)[24,25]. The presence of melanoma tumor nests between collagen bundles (Supplementary Fig. 6f) without an associated desmoplastic reaction, similar to the appearance of macrophages with tattoo pigment in Fig. 3c, supports that the cells are within pre-existing interstitial spaces and have not elaborated tissue-destructive digestive enzymes.

## Discussion

We demonstrate here that non-biological pigment particles (cosmetic tattoos and colloidal silver in the skin and endoscopically injected tattoo pigment in the colon) are found within subcutis and mesentery, respectively. These sites are distant from their sites of entry into the body and indicate movement across

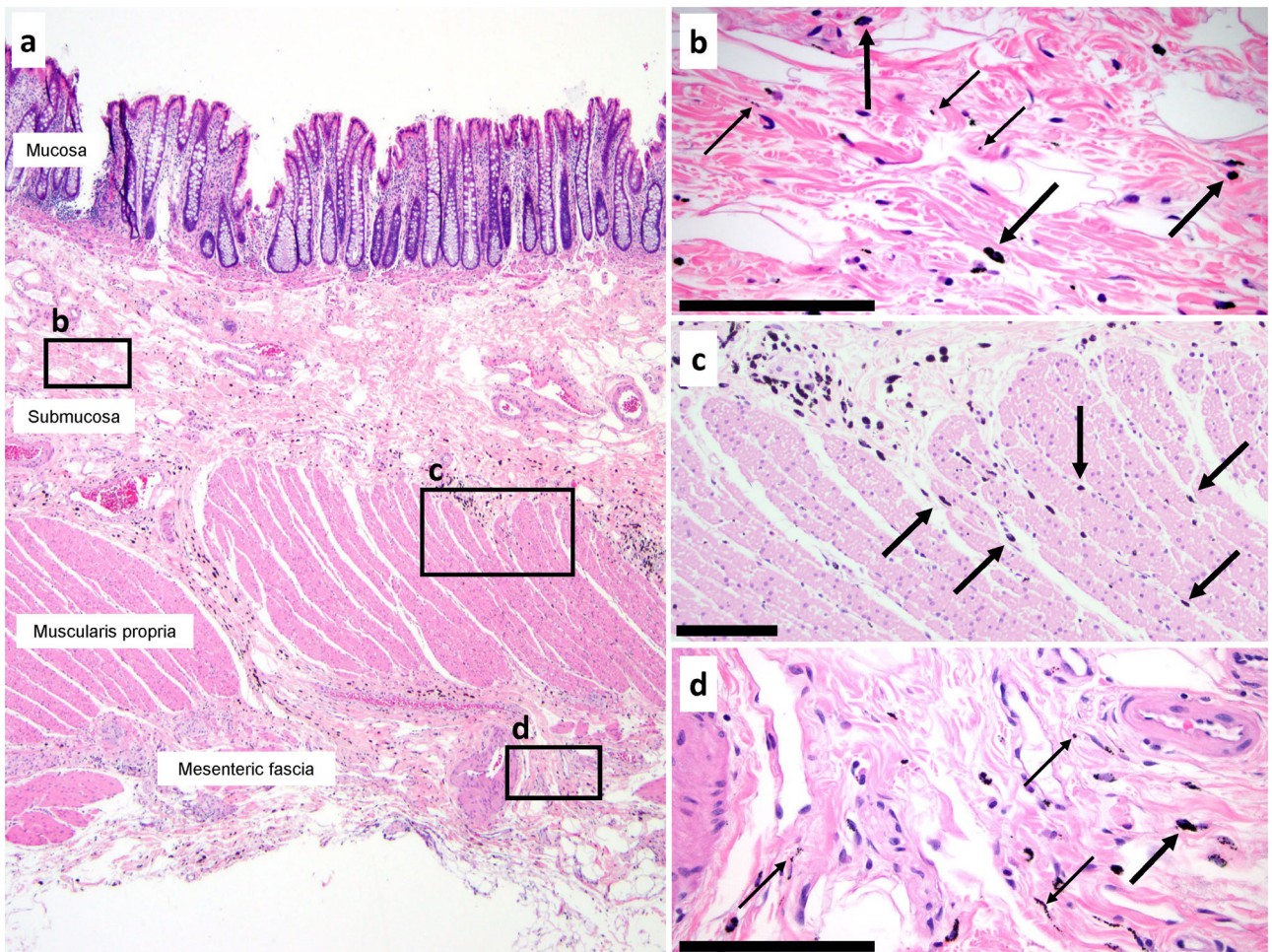

**Fig. 2 Tattoo pigment in interstitial spaces of the colon and mesenteric fascia. a** H&E section of colon resection specimen with endoscopically injected tattoo pigment. Pigment particles are present in submucosa, muscularis propria, and mesenteric fascia. **b** Higher magnification view of the submucosa demonstrates both intracellular particles (within macrophages; thick arrows) and extracellular particles (within interstitial spaces; thin arrows). **c** Intermediate power view of the muscularis propria shows pigment-containing macrophages (thick arrows) within interstitial spaces between muscle bundles (compare to Supplementary Figs. 6c, d for a similar display of movement by carcinoma). **d** Mesenteric fascia has intracellular and intercellular pigment as seen in all other layers. Scale bars = 100 μm.

classically defined tissue boundaries. We also use HABP staining to show that interstitial spaces are continuous between tissue compartments and fascial planes in the colon, skin, and liver, as well as within the fibrous tissues around blood vessels and nerves. Taken together, the data indicate that there is continuity of interstitial spaces within organs (shown here for skin and colon) and outside of organs (shown here for skin into subcutis and colon into mesentery).

Although the movement of pigment particles could occur via macrophage engulfment and migration, our data suggest that this is unlikely. The significantly smaller mean particle size in progressively deeper layers of the bowel wall suggests that particles were mostly carried via fluid flow, perhaps under peristaltic pressure, rather than by macrophage carriage and subsequent release, as cell-mediated transport would have resulted in an even distribution of sizes regardless of distance. The presence of colloidal silver through all layers of skin without evidence of macrophage engulfment also implies that cell-based particle movement is not needed (Supplementary Fig. 1). The argyria specimens also make it unlikely that the spread of tattoo pigment injection was due to mechanical pressure from the injection: the colloidal silver was applied as a topical agent on the skin, diffusing into the deeper layers without instrumentation. Its

dissemination into sub-epidermal stroma and stroma around adnexal structures highlights that basement membranes are permeable to the movement of minute extracellular particles (Supplementary Fig. 1).

The large interstitial spaces, which appear as empty white spaces on routine formalin-fixed, paraffin-embedded histochemically stained specimens, are filled with HA. This finding allowed us to demonstrate that interstitial spaces are continuous between tissue compartments and fascial planes in the colon, skin, and liver, as well as within the fibrous tissues around blood vessels and nerves, which may pass through multiple organs.

While this has not, to our knowledge, been demonstrated previously for skin and colon and their adjacent tissues, it confirms Mall's experimental demonstration of continuity within the portal tract stroma[1]. Mall demonstrated that pigmented gelatinous substances (cinnabar and Prussian blue) distributed into the microanatomic structures of the portal tract following their injection into the vascular supply of cat livers. Based on Mall's experimental observations, the perivascular and periductal portal tract stroma can be visualized as a unified network of fibers with intervening spaces that communicate with each other and are in continuity with the vascular (entrance) and lymphatic (egress) systems in the liver. Our findings are in agreement with Mall's

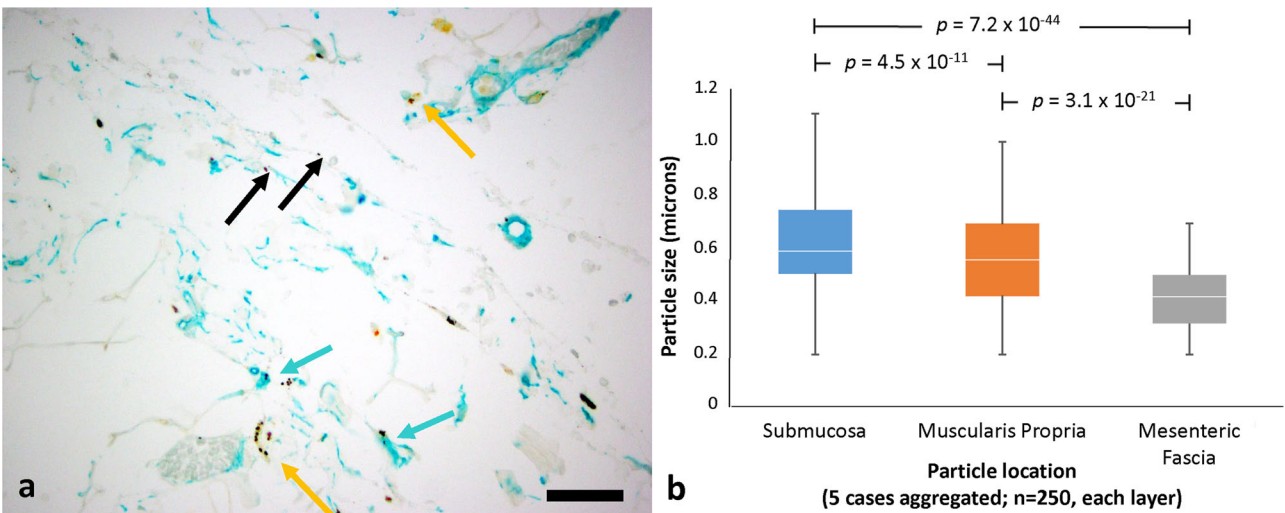

**Fig. 3 Tattoo pigment particle distribution in the colon. a** Pericolic mesenteric tissues with tattoo pigment within interstitial spaces (black arrows), within macrophages (yellow arrows), and within interstitial lining cells (teal arrows). Selected areas in the colonic submucosa, muscularis propria/superficial mesenteric fascia, and deep mesenteric fascia were examined at high power magnification. Double immunostain for CD68 (macrophage marker; yellow) and CD34 (interstitial lining cell marker; teal). Scale bar = 100 μm. **b** Distribution of particles by size in compartments at increasing distance from the lumen in aggregated data from all five colon tattoo specimens (50 particles sized for each of the three tissue layers per specimen, with a total of 250 particles represented per layer; each case presented separately in Supplementary Fig. 2). The boxes represent first, median, and third quartiles; the bars represent the range of data points.

observations: HA staining was present in a continuous fashion within portal tract stroma ensheathing hepatic arteries, portal veins, and the biliary tree, in support of there being a continuous space. The intrahepatic portal triads show how spaces in and around structures including arteries, veins, nerves, and ducts (such as ureters, urethra, epididymis, Fallopian tubes, salivary gland exocrine ducts) that pass through multiple tissues and organs could serve as conduits for interstitial fluid.

Particle movement by flow through fluid-filled channels likely arises from external and internal (physiologic) mechanical forces such as peristalsis of the gut, positional or mechanical pressure on the skin and subcutaneous tissues, and rhythmic compression of perivascular/adventitial stroma resulting from arterial wall expansion in systole. Diaphragmatic movement between inhalation and exhalation will also cause rhythmically oscillating pressures on tissues in the thorax and the abdomen, such as the mesentery, that may not have strong driving pressures from other sources. Movement of fluid through interstitial spaces of the perineurium may be subject to all these influences, as well as limited by the phenomenon of interstitial exclusion[21].

Conventional models suggest that the extracellular matrix serves as a physical barrier to cancer cell migration and that destructive breakdown by matrix metalloproteinases is a prerequisite step for cancer cell invasion and metastasis. However, extensive work has shown that cancer invasion, at least in its initial steps, is largely non-destructive, without significant tissue remodeling, and that malignant cells can traffic through pre-existing interstitial spaces, which likely serve as routes of least resistance[26–30]. In keeping with this work, our findings, as shown in Supplementary Fig. 6, may explain some features of classically recognized cancer behavior such as tumor cell spread within an organ tissue plane (single-cell filing of lobular carcinoma of the breast, linitis plastica of the stomach and, as shown in this study, the periductal spread of cholangiocarcinoma). Likewise, continuity across tissue planes could explain the clinically recognized "discontinuous" spread of cancer such as mesenteric tumor deposits of colorectal cancer and subcutaneous melanoma "in transit" metastases (Supplementary Fig. 6c–f).

These findings have important implications for other processes such as infections, autoimmunity, and host-microbiome interactions. Interstitial spaces across the human body could act as pathways for both commensal micro-organisms of the human microbiome and various pathogens. Direct continuity across tissue planes may explain necrotizing fasciitis, a fulminant form of soft tissue infection resulting from virulent bacterial strains that gain access to the interstitial spaces and cause widespread necrosis of subcutaneous and perimuscular fascia and even cross the blood:brain barrier to cause meningitis[31]. Continuity across the layers of the intestine and through the mesenteric and portal vein adventitia may provide a direct route for the translocation of gut bacteria to the liver (gut–liver axis)[32,33]. The existence of these pathways, in parallel with the mesenteric lymphatic and portal venous systems, may play a significant role in gut microbiota signaling not only in chronic liver disease and cirrhosis but also in the context of autoimmune disorders, both liver-specific (autoimmune hepatitis) and systemic[34]. New evidence shows that the central nervous system has a system of lymphatic vessels that line the dural sinuses and connect the cerebrospinal fluid space to the cervical lymph nodes; one anatomic route of this connection could be via the peri-arterial adventitia of the carotid arteries[35,36]. The presence of interconnected interstitial spaces of channels that track along the perineurium of the peripheral nerves provides a potentially novel route of communication between the gut and the brain (gut–brain axis)[37].

In conclusion, we confirm that the fibrous layers of nerves and blood vessels have no discrete separation from the fibrous components of the organs through which they travel and that, likewise, their interstitial spaces are not segregated from each other. Thus, we suggest that there is continuity of fibrous tissue interstitial spaces within and between organs and that these spaces are also potentially continuous between more distant parts of the body, at least along the vasculature and nerves. We speculate that these spaces serve as pathways for molecular signaling and cell trafficking in a way that is both in series and in parallel with the established pathways of the cardiovascular and lymphatic systems, although flow is likely limited by the structural proteins and

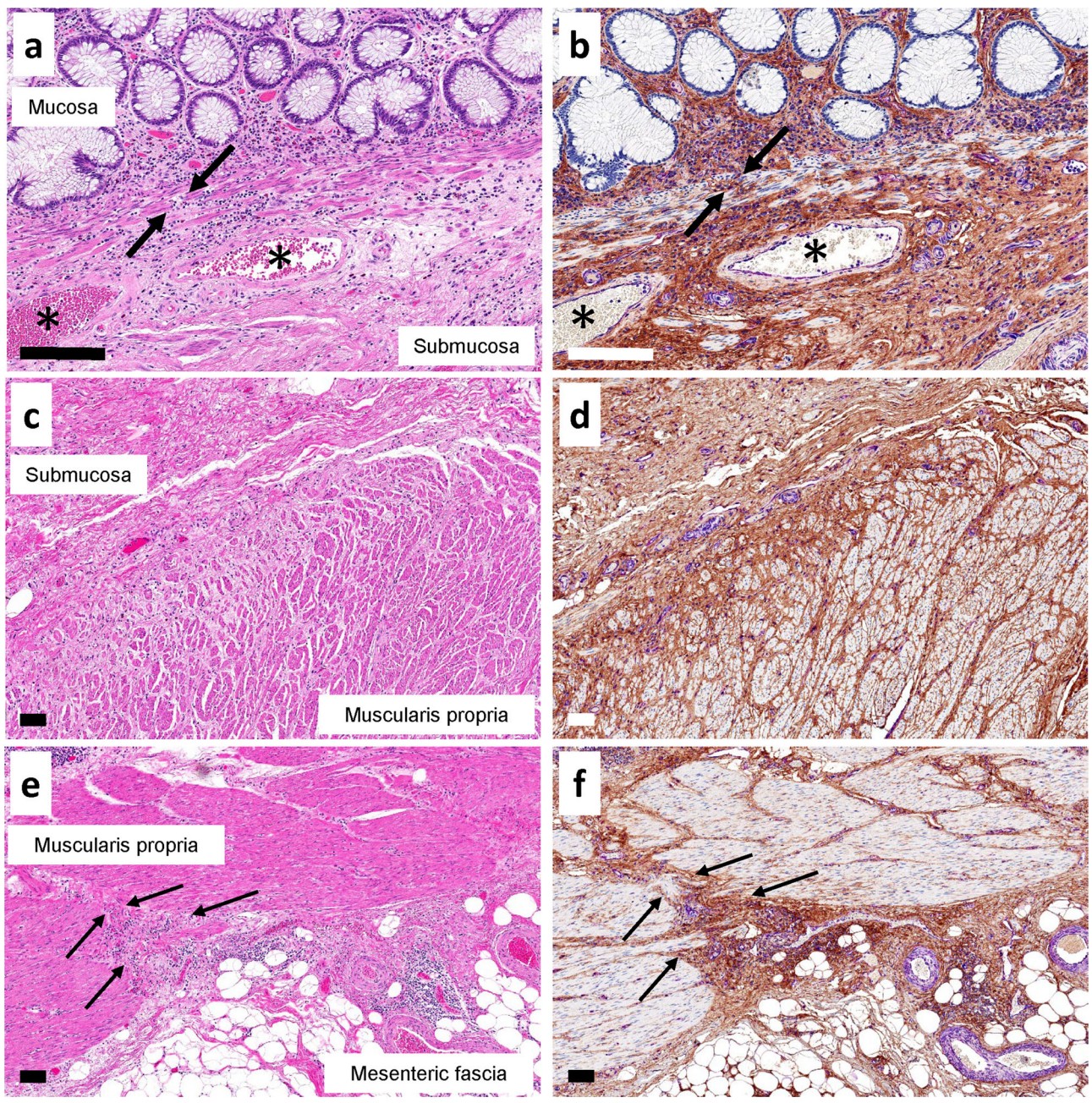

**Fig. 4 HA staining in interstitial spaces is continuous through all the layers of the colon into mesenteric soft tissues.** Adjacent colon samples were stained with H&E (left) and a triplex chromogenic assay (right) for HABP (brown), vimentin (magenta), and CD34 (teal). **a**, **b** Juncture of lamina propria to the submucosa, with channels observed through the muscularis mucosae (black arrows). There is continuity between all these layers by HA staining (right). * marks large veins in submucosa with HA staining showing continuity between perivascular stroma and the surrounding submucosal stroma. (×40 magnification). **c**, **d** Juncture between colonic submucosa and the muscularis propria. HA staining shows the continuity of HA-filled interstitial spaces from submucosa through small pericapillary channels between muscle bundles of the muscularis propria (right). (×10 magnification). **e**, **f** Edge of the fibrovascular bundle (arrows) passing between colonic wall, through muscularis propria, into subserosa and mesenteric fascia (right). In all multiplex-stained images, staining is as in Fig. 5. Scale bars = 100 µm.

glycosaminoglycans of the interstitium[21]. Given that the esti-mated volume of interstitial fluid in the body is more than three times the combined fluid volume of the cardiovascular and lymphatic systems[38], the existence of an interconnected inter-stitial compartment suggests an anatomic basis for understanding both physiological processes and disease pathophysiology.

## Methods

**Patients and tissue specimens**. Formalin-fixed, paraffin-embedded (FFPE) archival anatomic pathology tissue blocks were collected. Colon tissues were from surgical segmental colectomy specimens performed for the treatment of malignant polyps (five patients; Table 1), all of which contained tattoo pigment (India ink) that was injected according to standard techniques via a standard colonoscopy injection needle into the colonic submucosa adjacent to the lesion at the time of colonoscopy and prior to surgery. In each case, tissue from the tattoo site (middle of the specimen) and the normal distal or proximal surgical margin (whichever was furthest from the lesion) were examined for a total of ten specimens. Sections of normal skin were obtained from reduction mammoplasty specimens (five patients). Skin punch biopsy specimens were studied, including samples which included cosmetic tattoos (three patients) and two skin punch biopsy specimens containing colloidal silver in a single patient with argyria after topical skin application of colloidal silver. Sections of normal liver consisted of histologically normal liver

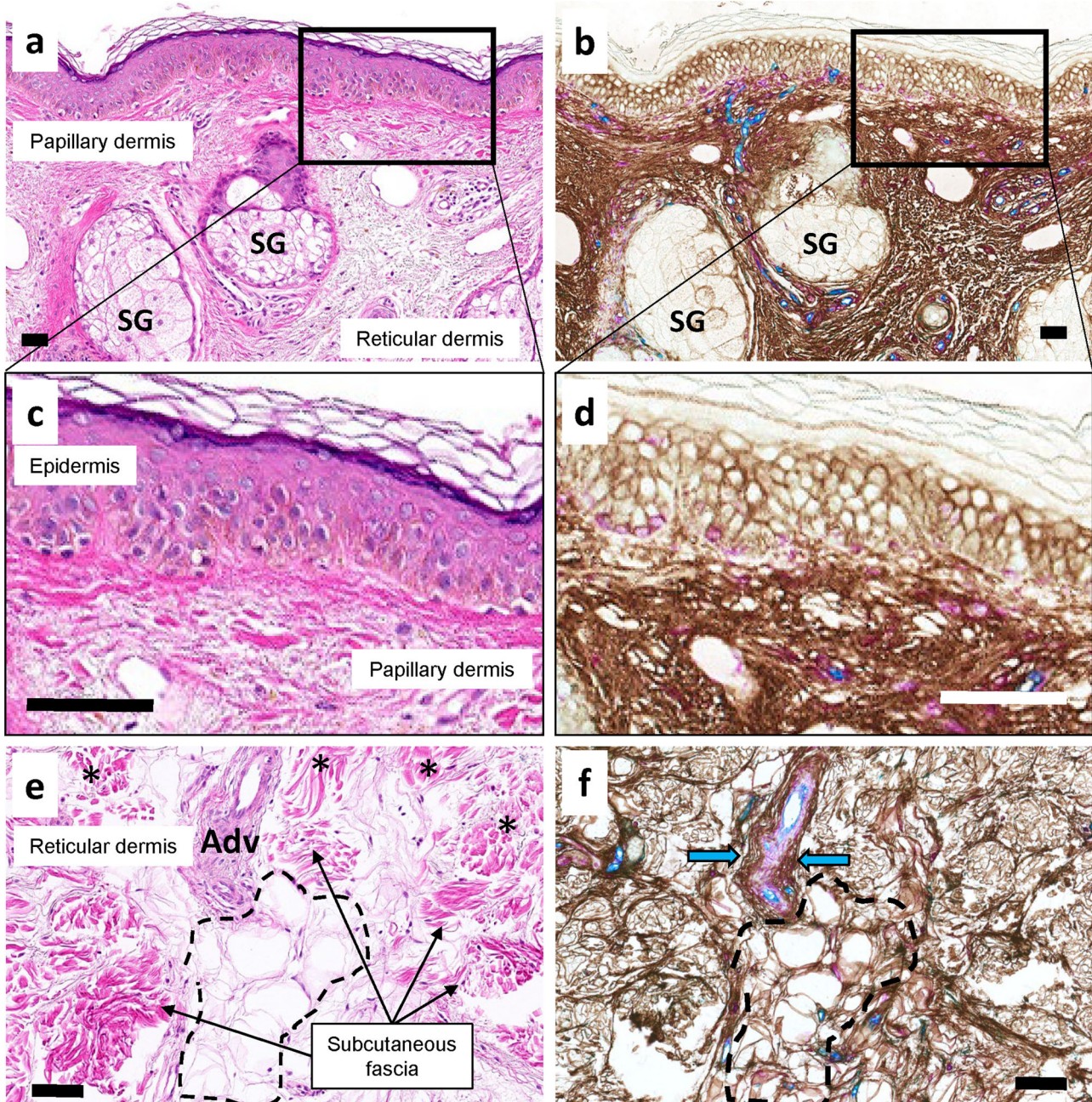

**Fig. 5 Continuity of interstitial spaces through all layers of the skin into the subcutaneous fascia.** Adjacent colon samples were stained with H&E (left) and a triplex chromogenic assay (right) for HABP (brown), vimentin (magenta), and CD34 (teal). **a, b** Junctional region between the papillary dermis and reticular dermis with adnexal sebaceous glands (SG). HABP staining of HA shows continuity from epidermal interstitial spaces through to the reticular dermis. (×10 magnification). **c, d** Rectangular regions from the top images expanded showing continuity from epidermal intercellular spaces into the papillary dermis (×60 magnification). **e, f** The lower reaches of papillary dermis (*) extending into the subcutaneous soft tissues with perivascular adventitia (Adv) and subcutaneous fascia separated by adipose tissue. HABP staining highlights the continuity between all these spaces including intercellular spaces and pericapillary spaces between adipocytes (dashed line) and the larger spaces of the reticular dermis, subcutaneous fascia, and perivascular adventitia (blue arrows). Scale bars = 100 μm.

tissue from resection specimens for metastatic tumor (four patients), focal nodular hyperplasia (two patients), cavernous hemangioma (one patient), and a wedge liver biopsy taken during cholecystectomy (one patient). Additional inclusion criteria were: bowel specimens—the presence of all anatomic compartments (mucosa, submucosa, subserosa, and mesentery) within the tissue section; skin specimens—the presence of epidermis, dermis, and subcutaneous fascia and adipose tissue. Samples from patients <18 years old were excluded.

The study was conducted in accordance with the guidelines and regulations and with the approval of the New York University Langone Health Institutional Review Board (ID# i18-01106, renewed 8/20/2020). Under these guidelines, since the study tissues consist of retrospectively retrieved, archival tissues and selected relevant

clinical data that were de-identified prior to experimental procedures and analyses, informed consent was waived.

**H&E staining, scanning, and decolorization.** FFPE specimens were sectioned at 5 μm onto charged slides (Fisher Scientific, Cat # 22-042-924). Slides were dried for 1 h at 60 °C, deparaffinized in xylene, rehydrated through a graded series of ethanols, and rinsed in distilled water. Slides were hematoxylin (Richard-Allan Scientific, Cat# 7211) and eosin (Leica, Cat# 3801619) stained using standard laboratory protocol[39]. Upon completion of staining, slides were dehydrated through a series of ethanols and xylene, and mounted with Cytoseal 60 (Richard-Allan Scientific, Cat# 8310-4). Slides were

**Table 1 Pathologic data for colon resection specimens.**

| Case # | Anatomic location of tumor | Time interval between endoscopic tattoo injection and colorectal resection (days) | Anatomic location in resection specimen of sampled normal tissue in relationship to tumor (proximal vs distal) | Distance of sampled normal tissue from tumor (cm) |
|---|---|---|---|---|
| 1 | Descending colon | 82 | Distal | 39.0 |
| 2 | Rectosigmoid colon | 21 | Proximal | 10.5 |
| 3 | Sigmoid colon | 47 | Proximal | 9.5 |
| 4 | Sigmoid colon | 22 | Proximal | 15.1 |
| 5 | Rectum | 53 | Proximal | 10.3 |

scanned using a Leica Biosystems Aperio AT2 System and digitally archived via eSlide Manager (Version 12.3.2.5030). Following scanning, H&E slides were immersed in xylene to remove coverslips. Slides were rehydrated through xylene, graded ethanol, and running distilled water, decolorized with 10% acetic acid in 70% ethanol for 1 h, rinsed in distilled water, and then further decolorized in 70% ethanol for 2–3 h. Evaluation for the decolorization end-point was checked every 30 min[40,41]. Once decolorization was complete, slides were rinsed in running distilled water.

**Multiplex immunohistochemistry.** Unconjugated murine anti-human Vimentin (Ventana Medical Systems, Cat# 790-2917, RRID: AB_2335925) clone V9, unconjugated murine anti-human CD34 (Ventana Medical Systems, Cat# 790-2927, RRID: AB_2336013) clone QBEnd/10, and unconjugated murine anti-human CD68 (Ventana Medical Systems, Cat# 790-2931, RRID: AB_2335972) clone KP1 were used for chromogenic immunohistochemistry. Biotinylated HABP (Calbiochem, Cat# 385911) was used for a non-immune chromogenic assay. The protein binds specifically to HA (≥2000 M.W)[42]. Biotinylated HABP is directly detected using a streptavidin peroxidase-DAB detection system.

Chromogenic immunohistochemical multiplexing (mIHC) was performed on a Ventana Medical Systems Discovery Ultra using Ventana reagents except as noted, according to the manufacturer's instructions and best practices[43,44]. Slides were dried in a 60 °C incubator for 1 h and deparaffinized on-instrument. Decolorized samples for mIHC bypass prerun incubation and deparaffinization and are started directly from buffer. All sample sets were run with a tissue microarray as positive, negative, and mIHC crossover controls.

For the HABP–Vimentin–CD34 triplex assay, endogenous peroxidase was blocked with 3% hydrogen peroxide for 8 min at 37 °C. HABP was applied at 0.5 µg/ml (1:100 dilution) in tris-buffered saline with 1% bovine serum albumin (TBSA) and incubated for 12 h at room temperature. The biotinylated protein was then directly detected using horseradish peroxidase-conjugated streptavidin with DAB substrate. Sections were then antigen retrieved using Cell Conditioner 1 (Tris-Borate-EDTA ph8.5) for 20 min at 95 °C. Peroxide blocker was reapplied as above. Vimentin antibody was applied without dilution and incubated for 20 min at 37 °C followed by a goat anti-mouse horseradish peroxidase (HRP)-conjugated multimer applied for 8 min at 37 °C. This was detected with purple (tyramide-TAMRA) chromogen for 8 min at 37 °C. Subsequently, sections were denatured in Reaction Buffer (Cat# 950-300) for 32 min at 95 °C. Endogenous peroxidase was blocked as previously described. CD34 was applied neat for 60 min at 37 °C. A goat anti-mouse secondary HRP-conjugated multimer was applied neat for 8 min at 37 °C and detected using teal (tyramide-Cy5) chromogen for 16 min at 37 °C. Slides were subsequently dehydrated, coverslipped, and scanned.

Extracellular versus intracellular localization of pigment particles was assessed on sections stained with duplex immunohistochemistry for CD68 (macrophage marker) and CD34 (a marker of interstitial lining cells). For the CD68–CD34 duplex assay, the CD34 antibody was applied neat to deparaffinized slides for 60 min at 37 °C. Goat anti-mouse secondary HRP-conjugated multimer secondary was applied neat for 8 min at 37 °C and detected using Teal chromogen for 8 min at 37 °C. Slides were denatured in reaction buffer for 32 min at 95 °C followed by antigen retrieval using Cell Conditioner 1 for 20 min at 91 °C. CD68 was applied neat for 32 min at 37 °C. A goat anti-mouse secondary alkaline phosphatase-conjugated multimer was applied for 8 min at 37 °C and detection was completed using Yellow (tyramide-dabsyl) chromogen for 8 min at 37 °C. Slides were subsequently dehydrated, coverslipped, and scanned.

**Particle size measurement.** In each of the five tattooed colon specimens, 50 tattoo pigment particles were measured within each layer: submucosa, muscularis propria, and mesenteric fascia. The eSlide Manager digital annotation ruler tool was used. It was not possible to measure the size of particles within macrophages because particles were often stacked upon each other in the cytoplasm. Student $t$-test was used to analyze discrete data.

**Statistical analysis and reproducibility.** All statistical analyses were two-tailed. $p$-values < 0.05 were considered statistically significant.

**Reporting summary.** Further information on research design is available in the Nature Research Reporting Summary linked to this article.

## Data availability
The data that support the findings of this study are available from the corresponding author upon reasonable request.

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

## Acknowledgements

The NYULH Center for Biospecimen Research and Development, Histology and Immunohistochemistry Laboratory (RRID:SCR_018304) is supported in part by the Laura and Isaac Perlmutter Cancer Center Support Grant: NIH/NCI P30CA016087 and the National Institutes of Health S10 Grants NIH/ORIP S10OD01058 and S10018338. This work wassupported by the Center for Engineering MechanoBiology (CEMB), an NSF Science and Technology Center, under grant agreement CMMI: 15-48571.

## Author contributions

O.C. contributed to study design, was responsible for the selection of study tissues, all microscopic analyses, photomicroscopy, figure creation, and co-wrote the manuscript. D.H.R.A. carried out particle size measurements and statistical analysis. R.I., J.L., C.Z.L., and Y.N.P. contributed to study design, interpreting data, and editing manuscript. L.M.D. contributed to tissue staining. L.C. and B.Z. contributed to study design, performed all tissue staining, and co-wrote the "Methods" section. R.G.W. and N.D.T. conceived the project, contributed to study design, analyzed data, co-wrote the manuscript, participated in figure design, and oversaw the project. R.G.W. and N.D.T. contributed equally.

## Competing interests

The authors declare no competing interests.
