## [Peer Review File · Communications Biology]

Reviewers' comments:

Reviewer #1 (Remarks to the Author):

Manuscript is well written. It is overall clear. Introduction is relevant and provides sufficient information to understand the rationale of the study. Methods are generally appropriate. Procedural details are described adequately. Results are overall clear.

Evidence from study does provide information that to some extent in localized areas communication and fluid flow exists.

Sample size seems to be small to actually provide a strong evidence to support the hypothesis.

The evidence may also be questionable as evidence of fluid flow is being described in an altered microenvironment as certain specimens are derived from malignant samples.

Details of the statistical analysis of study and its interpretation has not been provided.

Some issues with the article have been highlighted and details have been provided in the attachment file containing review.

Reviewer #2 (Remarks to the Author):

Thank you very much for the opportunity to review a very interesting article entitled "Interstitial spaces are continuous across tissue and organ boundaries in humans". Knowledge of this subject seems to be essential in understanding, for example, the spread of cancer. The article deals with a very important "science section" and is eligible for publication in Nature Biology Communications with some revisions:

Abstract:

The abbreviation IS has not been expanded.

I also miss a detailed description of injection techniques?

Introduction: OK

Where is the Material and Methods section?

Results: There are the M&M delicately mentioned here, however I believe that an M&M section should be added and the injection method carefully described, step by step, make a good test protocol that will serve next scientists as an example (thus also achieving high citation).

What are the conclusions? please specify more.

Point-by-Point Responses to Reviewers (Cenaj et al, COMMSBIO-20-2236-T)

Reviewer #1

Manuscript is well written. It is overall clear. Introduction is relevant and provides sufficient information to understand the rationale of the study. Methods are generally appropriate. Procedural details are described adequately. Results are overall clear. Evidence from study does provide information that to some extent in localized areas communication and fluid flow exists.

Sample size seems to be small to actually provide a strong evidence to support the hypothesis.

We have increased the sample sizes for colon, liver, and skin. It should be noted that using clinical tissue limits the degree to which normal tissues are available, particularly since autopsy tissues do not allow for the hyaluronic acid staining performed in the study. We have compensated, where possible, as noted:

1. There is essentially no reason to ever resect a normal colon, but we have obtained the normal tissue surgical margins most distant from each of the 5 cancer/tattoo resections in the study. This brings the total number of colon specimens studied to 10. Data for distance of these specimens from the tumor/tattoo site is included a new Table 1.

2. Normal skin is routinely part of most reduction mammoplasty specimens and these are also known to not have breast cancer in them. Five additional such specimens have been added.

3. Five additional normal livers from clinically obtained specimens have been included. These were deemed histologically normal or from non-lesional liver resected along with benign developmental lesions (focal nodular hyperplasia, hemangioma) or metastatic carcinoma in which the metastasis was small and abundant normal liver was present at the margin of resection.

The opening paragraph of Methods has been rewritten as follows:

"Formalin-fixed, paraffin-embedded (FFPE) archival anatomic pathology tissue blocks were collected. Colon tissues were from surgical segmental colectomy specimens performed for the treatment of malignant polyps (5 patients; see Table 1 for details), all of which contained tattoo pigment (India ink) that was injected via a standard colonoscopy injection needle into the colonic submucosa adjacent to the lesion at the time of colonoscopy and prior to surgery. In each case, tissue from the tattoo site (middle of the specimen) and the normal distal or proximal surgical margin (whichever was furthest from the lesion) were examined for a total of 10 specimens. Sections of normal skin were obtained from reduction mammoplasty specimens (5 patients). Skin punch biopsy specimens were studied, including samples which included cosmetic tattoos (3 patients) and two skin punch biopsy specimens containing colloidal silver in a single patient with argyria after topical skin application of colloidal silver. Sections of normal liver consisted histologically normal liver tissue from resection specimens for metastatic tumor (4 patients), focal nodular hyperplasia (2 patients), cavernous hemangioma (1 patient), and a wedge liver biopsy taken during cholecystectomy (1 patient). Additional inclusion criteria were: bowel specimens – the presence of all anatomic compartments (mucosa, submucosa, subserosa and mesentery) within the tissue section; skin specimens – the presence of epidermis, dermis, and subcutaneous fascia and adipose tissue. Samples from patients less than 18 years old were excluded."

The evidence may also be questionable as evidence of fluid flow is being described in an altered microenvironment as certain specimens are derived from malignant samples.

As noted above, normal colon tissue has been obtained significant distances away from the resected tumor (see new Table 1 for measurements from tumor and locations).

Details of the statistical analysis of study and its interpretation has not been provided.

Statistical analysis of the quantitative particle size data is described in the methods section, page 13.

Additional Comments:

1. Abstract Line 26: “We recently described fluid flow through such human fibrous tissues.” Comment: Instead of “We recently described” kindly write “We validate/corroborate”..... As the finding is not novel but this fluid flow is already known to exist.

Response: Changed to "we validated"

2. Introduction Line 56: “We recently described fluid flow through large interstitial spaces of the human extrahepatic bile duct submucosa and the human dermis, 50-70 μm below the epithelial surface.” Comment: If this is a reference statement please rewrite and change to “Studies have” instead of “We recently”.

Response: Done.

3. Lines 57-63 “We further showed that other fibrous tissues, including the submucosae of all other visceral organs and the subcutaneous fascia, are structurally similar, and hypothesized that they likewise support fluid flow. In all of these tissues, the spaces were defined by a network of collagen bundles 20-70 μm in diameter. Many of the collagen bundles were lined by spindle-shaped cells that co-expressed vimentin and CD34, but were devoid of endothelial ultrastructural features and were thus considered fibroblast-like cells. In this context, we refer to them as "interstitial lining cells." Comment: Are these author’s findings? If yes are they correctly placed in introduction?

This a summary of prior work which led to the current research. It is therefore placed in the introduction to provide necessary context for performing this current study.

4. Lines 74-79 “We demonstrate continuity across organ boundaries and between spaces in all fibrous tissues studied, including the perineurium and vascular adventitia within them. We suggest that there is a broad and interconnected network of interstitial fluid-filled channels throughout the body, including the structural coverings of nerves and vessels, and that this has significant implications for molecular signaling, cell trafficking, and the spread of malignant and infectious disease.” Comment: Not clear whether these are the author’s results and discussion. If so are they correctly placed under introduction?

Deleted from the final paragraph of the introduction and the remainder of that paragraph has been rewritten accordingly.

5. Methods Lines 237-240 : “Formalin-fixed, paraffin-embedded (FFPE) archival anatomic pathology tissue blocks were collected from 238 a) surgical segmental colectomy specimens performed for the treatment of malignant polyps (5 239 patients), all of which contained tattoo pigment (India ink) that was injected into colonic submucosa 240 adjacent to the lesion at the time of colonoscopy and prior to surgery”

Comments: 1. It would be helpful to provide details of time since injection of pigment in colonic submucosa and retrieval of tissue sample; time since tattoo and biopsy specimen etc. and details of how the specimens were retrieved.

We are appreciative of this excellent idea. We have obtained these data. Collection method is included in the Methods and they are included in the results section and in the new Table 1.

Table 1. Pathologic data for colon resection specimens.

Case #	Anatomic location of tumor	Time interval between endoscopic tattoo injection and colo-rectal resection (days)	Anatomic location in resection specimen of sampled normal tissue in relationship to tumor (proximal versus distal)	Distance of sampled normal tissue from tumor (cm)
1	Descending colon	82	Distal	39.0
2	Rectosigmoid colon	21	Proximal	10.5
3	Sigmoid colon	47	Proximal	9.5
4	Sigmoid colon	22	Proximal	15.1
5	Rectum	53	Proximal	10.3

2. Is the selection of malignant tissue justified and correct in substantiating the presence of interstitial fluid flow? Would it be more relevant if study had explained interstitial flow using biological/ non biological particles in non-malignant specimens? Presence of malignancy may confound the study results as normal interstitium differs from that in

tumorous tissues. Tumor microenvironment and microcirculation is different as compared to normal tissues.

We have gone back to the original colon cases and selected normal tissue from each, either the distal or the proximal surgical margin of each colectomy case, whichever was further. The location compared to the tumor site and distances from the tumor site are included in the new Table 1. These are described in the results and in the discussion.

6. Lines 247-249: "The study was conducted in accordance with the guidelines and regulations and with the approval of the 248 New York University Langone Health Institutional Review Board and, under those guidelines, informed 249 consent was waived." Comments: Provide confirmation about ethical clearance for the study.

We have further specified the ethical clearance for the study:

Page 11: "The study was conducted in accordance with the guidelines and regulations and with the approval of the New York University Langone Health Institutional Review Board (#i18-01106, renewed 8/20/2020). Under these guidelines, since the study tissues consist of retrospectively retrieved, archival tissues and selected relevant clinical data that were de-identified prior to experimental procedures and analyses, informed consent was waived."

7. Lines 303-306: "Sizes of 50 tattoo pigment particles were measured using eSlide Manager digital annotation ruler tool. 304 It was not possible to measure the size of particles within macrophages because particles were often 305 stacked up on each other in the cytoplasm. Student t test was used to analyze discrete data. All 306 statistical analyses were two-tailed. P values less than 0.05 were considered statistically significant." Comments Kindly provide details of the number of the tattoo pigment particles that were finally measured as per their location and provide details of statistical analysis.

The methods of particle measurement and details of the statistical analysis are provided in the Methods, page 13. We have returned to all tattoo cases and measured all of them: 50 particles x 3 layers x 5 cases = 750 particles in total. The data for all cases in aggregate are presented in a revised Figure 4B. The data for each individual case are provided in a new Supplementary Figure 1.

8. References Line 427: “van der Waal” Comment Van should be spelled with V in capital letters.

V now capitalized in reference 7.

9. Line 430: “Stecco, C. Functional Atlas of the Human Fascial System, (Churchill Livingstone, London, 2018).”

Comment Please provide the page number

Pages 1-2 have been added to reference 8.

10. Line 488-489: “Carson, F. & Cappellano, C. Histotechnology: A Self Instructional Text, (ASCP Press, Chicago, 2015).

pp 114-123 have been added to reference 35

Reviewer #2

Thank you very much for the opportunity to review a very interesting article entitled "Interstitial spaces are continuous across tissue and organ boundaries in humans". Knowledge of this subject seems to be essential in understanding, for example, the spread of cancer. The article deals with a very important "science section" and is eligible for publication in Nature Biology Communications with some revisions.

Abstract:

The abbreviation IS has not been expanded.

Done. "..._continuity of interstitial spaces"

I also miss a detailed description of injection techniques?

This has now been added to the Methods.

Where is the Material and Methods section?

As per Communications Biology style, the "Methods" section comes at the end of the text, before the figure legends, pages 10-13.

Results: There are the M&M delicately mentioned here, however I believe that an M&M section should be added and the injection method carefully described, step by step, make a good test protocol that will serve next scientists as an example (thus also achieving high citation).

This is in accord with the house style for Results in Communications Biology. As noted above, full Methods are provided later, at the end of the body of the manuscript text, before the figure legends.

What are the conclusions? please specify more.

Conclusions are present throughout the Discussion, but the most important summarized in the final paragraph of this section, on page 10. We have added "In conclusion..." to highlight them better.

REVIEWERS' COMMENTS:

Reviewer #1 (Remarks to the Author):

The manuscript is well written and clear. Most of the queries that were raised, have been addressed satisfactorily by the authors. Though the sample size in the study is small and study material is derived from malignant samples (altered microenvironment) which are not a true representation of a normal population sample. Manuscript may have a potential for being accepted due to the scientific soundness of the evidence provided from the study.